# Climate Change Impact on the Cultural Heritage Sites in the European Part of Russia over the Past 60 Years

Elena Vyshkvarkova * and Olga Sukhonos

Institute of Natural and Technical Systems, Sevastopol 299011, Russia
* Correspondence: aveiro_7@mail.ru

**Abstract:** Climate change is causing damage to infrastructure, ecosystems, and social systems, including cultural heritage sites. In the European part of Russia, there are 20 UNESCO-listed cultural heritage sites situated in different climatic conditions. This study assesses the impact of climate change on these sites by using ERA5 re-analysis data to calculate two frost damage indices and two salt weathering indices for the period 1960–2020. The findings indicate a rise in frost damage and salt weathering at cultural heritage sites in northern Europe, primarily due to changes in air temperature and water in the atmosphere, which are the main parameters responsible for the destruction of stone and brick structures. Given the observed and predicted trends in the main meteorological parameters, the detrimental destructive impact of climate change on cultural heritage sites will only increase. In view of the significant length of Russia from north to south and the difference in climatic conditions, measures for the adaptation and protection of cultural heritage sites must be adapted to local conditions and consider the material from which the object is made.

**Keywords:** climate change; cultural heritage; European part of Russia; frost damage; salt weathering



## 1. Introduction

Climate change has become a serious problem for mankind due to adverse environmental and social consequences. Over the past 50 years, the mean surface air temperature has increased at an unprecedented rate [1]. Climate change is affecting all aspects of human and natural systems, including both natural and cultural World Heritage sites. Protecting, securing and building sustainable management of these priceless sites should be a cross-government priority. Assessing the impact of climate change on the world's cultural heritage must consider the complex interactions between natural, cultural and social systems.

Cultural heritage sites, historical buildings and natural sites are constantly exposed to the environment and change under its influence. The progressive loss of historical and archaeological sites due to climate change is often overlooked [1]. Climate change poses an additional potential threat, accelerating the rate of degradation. This is due to the strengthening of physical, chemical and biological mechanisms that cause the destruction of materials. According to IPCC experts [1], climate change affects the frequency and intensity of hazardous events such as droughts, floods and landslides, with inevitable wide-ranging consequences, including for cultural heritage. Climate change poses a serious threat to the preservation of cultural heritage in Europe, both tangible and intangible [2–4].

Studies to assess the impact of climate change on cultural heritage sites have been and are being carried out within the framework of European projects and organizations, such as NOAH's ARK Global Climate Change Impact on Built Heritage and Cultural Landscapes, Climate for Culture, ICOMOS, UNESCO and organizations of national committees.

There are many impacts of climate change affecting cultural heritage, for example changing precipitation patterns, increased occurrence of extreme events such as droughts and floods and rising sea levels [5–8]. Changes in groundwater levels, hydrological cycle,

soil temperature, crystallization and dissolution of salts, more frequent intense precipitation leading to erosion will seriously damage cultural heritage sites.

Crystallization and dissolution of salts caused by wetting and drying damage walls, frescoes, wall paintings of archaeological sites and other surfaces [9,10]. Salt weathering is one of the most important criteria for the degradation of historical heritage sites, due to phase changes in relative humidity [11]. This damage occurs during the cycles of crystallization and salt dissolution.

UNESCO cultural heritage sites located in the coastal zone are subject to destruction due to sea level rise, which has been observed in recent decades due to global warming [12–16]. Sea level rise and coastal erosion caused by storms endanger coastal archaeological sites and heritage sites around the world [17–19]. An increased risk of landslides causes either the loss of structures located on slopes, or the filling and damage of structures by stones, mud and debris [9,10]. An increase in the risk of landslides has been found across Europe [20], in particular in the UK [21]. Extreme events such as droughts, heatwaves can increase the risk of ignition and fire spread [9,22,23].

In the territory of Russia there are 30 sites included in the World Heritage List, of which 19 are sites of tangible cultural heritage [24]. For the territory of Russia, the problem of climate change is acute. The average warming rate on the territory of Russia for the period 1976–2021 was 0.49C/10 years [25]. The maximum summer warming is observed in the south of the European part of Russia (0.74C/10 years). According to Roshydromet data [24], the trend in annual precipitation for the period 1976–2021 is averaged for the territory of Russia, positive and amounts to 0.8%/10 years. A decrease in annual precipitation (at a rate of no more than 6% of the norm for 10 years) was found in the European part of Russia in the latitudinal zone 50–60° N. In the central part and in the south of the European part of Russia, the trend towards a decrease in the amount of summer precipitation continues. There is an increase in climate extremes throughout the country. The mentioned changes certainly affect the building materials of cultural heritage sites in Russia, destroying them [26,27]. Due to wind influences and seasonal changes in the moisture content of the logs, there was a decrease in the friction forces between the logs, which ensure the stability of the construction of the Kizhi Pogost [28]. Against the backdrop of global warming and the associated sea level rise in St. Petersburg, 135 sites of historical and cultural significance are currently subject to flooding [26]. Intensive climate changes, such as rainfall, thaws, as a result of changes in the dynamics and mechanism of formation of waste and groundwater—all this leads to an increase in the impact of dangerous exogenous geological processes on the soils of the foundations of cultural heritage sites. As a result of the development of abrasion processes on the territory of the Assumption Cathedral and Monastery of the town-island of Sviyazhsk, cracks and slopes of the walls in the direction of the slope were noted in the walls enclosing the territory of the monastery [29]. The aim of the article is to study a climate change impact on cultural heritage sites in the European part of Russia. Observed and predicted trends in key meteorological parameters have and will continue to have devastating impacts on cultural heritage sites.

## 2. Materials and Methods

### 2.1. Cultural Heritage Sites in the European Part of Russia

There are 19 UNESCO cultural heritage sites in the European part of Russia [24] (Figure 1):

1. Historic Centre of Saint Petersburg and Related Groups of Monuments (Saint Petersburg, XVIII–XX centuries)
2. Kizhi Pogost (Lake Onega, in Karelia, XVIII–XIX centuries)
3. Kremlin and Red Square (Moscow, XIII–XVII centuries)
4. Cultural and Historic Ensemble of the Solovetsky Islands (Arkhangelsk region, XVI–XVII centuries)
5. Historic Monuments of Novgorod and Surroundings (Novgorod, XI–XVII centuries)
6. White Monuments of Vladimir and Suzdal (Vladimir and Suzdal, XII–XIII centuries)

7. Architectural Ensemble of the Trinity Sergius Lavra in Sergiev Posad (Moscow region, Sergiev Posad city, XV–XVIII centuries)
8. Church of the Ascension, Kolomenskoye (Moscow, XVI century)
9. Ensemble of the Ferapontov Monastery (Vologda region, the village of Ferapontovo, XV-XVII centuries)
10. Historic and Architectural Complex of the Kazan Kremlin (Republic of Tatarstan, Kazan, XVI–XXI centuries)
11. Citadel, Ancient City and Fortress Buildings of Derbent (Republic of Dagestan, VI-XIX centuries)
12. Ensemble of the Novodevichy Convent (city of Moscow, XVI–XVII centuries)
13. Historical Centre of the City of Yaroslavl (Yaroslavl region, city of Yaroslavl, XVI–XX centuries)
14. Bolgar Historical and Archaeological Complex (Republic of Tatarstan, X-XV centuries)
15. Assumption Cathedral and Monastery of the town-island of Sviyazhsk (Republic of Tatarstan, Sviyazhsk, from the 16th century)
16. Churches of the Pskov School of Architecture (Pskov region, city of Pskov, from the 12th century)
17. Petroglyphs of Lake Onega and the White Sea (Republic of Karelia, 4–5 thousand years BC)
18. Struve Geodetic Arc
19. Curonian Spit

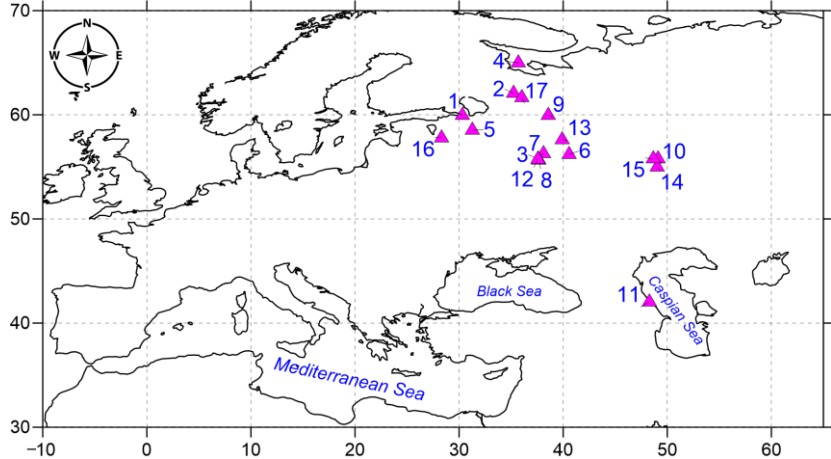

**Figure 1.** Location of the cultural heritage sites in the European part of Russia (the numbers in the figure correspond to the numbering of sites in the text above).

Most of the cultural heritage sites in the European part of Russia are made of stone, limestone, marble, local shell rock, and red brick. The Struve Arc and Curonian Spit were not considered because they represent disparate station points in the case of the Struve Arc and sand dune peninsula (Curonian Spit).

In this work, we have focused on UNESCO-listed cultural heritage sites, as they usually receive little attention, and the damage caused by climate change usually falls outside the attention of researchers [1,30]. The length of Russia, the diversity of the climate, the severe conditions in the Far North, the location of sites on the coasts of the seas and lakes (Section 2.2) suggests an aggressive impact and intensification of destruction, especially in the context of the observed unprecedented changes in temperature and humidity [1].

*2.2. Climatic Characteristics of the Locations of Cultural Heritage Sites*

The cultural heritage sites are located in contrasting climatic zones. Each site has unique features—the time of construction, the material from which it is made, the climatic conditions in which it is located, which requires an individual approach to assessing the

impact of climatic conditions on the degradation of materials. Sites made of the same building material, being in contrasting climatic conditions (for example, the far north and southern regions of Russia), have their own characteristics of destruction. Below is a description of the climatic conditions of the areas from north to south in which cultural heritage sites are located.

The northernmost site is located in the Far North of the Russian Federation—Cultural and Historic Ensemble of the Solovetsky Islands, located in the Arkhangelsk Region. The climate of the Solovetsky archipelago is marine, with a transition to continental. The climate on the islands is temperate and is determined by the location of the archipelago in the polar latitudes and the surroundings of the sea.

The architectural ensemble of the Kizhi Pogost (Republic of Karelia) is located in a temperate continental climate with maritime features. The average annual air temperatures in Zaonezhie is positive and amounts to 2.3 °C. A feature of the atmospheric circulation over Karelia is the so-called western transport (zonal circulation), often interrupted by intrusions of cold air from the Arctic.

The ensemble of the Ferapontov Monastery in the Vologda region and the historical center of St. Petersburg are located at the same latitude, but the distance from the sea creates differences in the climatic conditions of the two regions. Vologda is located in the temperate continental climate zone. The weather is unstable: thaws are observed in winter, severe frosts are possible in spring. The climate of St. Petersburg is moderate and humid, transitional from continental to maritime. This region is characterized by a frequent change of air masses, largely due to cyclonic activity.

The climate of Veliky Novgorod is temperate continental, with cold snowy winters and moderately warm summers. Winter lasts from mid-November to early April, its average temperature is −4 °C, the temperature often falls below −15 °C, usually in late January—early February. The climate of Pskov is transitional from temperate maritime to temperate continental, with mild winters and warm summers. The historical center of Yaroslavl is located in the temperate continental climate zone. The moderating influence of the Atlantic Ocean provides a relatively small fluctuation in seasonal temperatures.

The sites located in Moscow, the Republic of Tatarstan and in Suzdal and Vladimir are located in the same latitudinal zone. The climate is temperate continental with a pronounced seasonality. The southernmost site of cultural heritage is located in Dagestan in the city of Derbent. The climate of Derbent is transitional from temperate to subtropical semi-dry. The climate is affected by the Caspian Sea.

*2.3. Data*

The data of the main meteorological parameters (surface air temperature, precipitation, dew point to calculate relative air humidity) from the ERA5 reanalysis were used. We used daily data with a spatial resolution of 0.5*0.5° for the territory of the European part of Russia. The study period is 1961–2020, which includes the last two climatic periods (30-year periods recommended by the WMO for calculating climate averages).

*2.4. Methods*

Air temperature and water in the atmosphere are fundamental parameters influencing cultural heritage sites, since they significantly affect the durability of materials. Ancient structures were created in certain local climatic conditions and have high porosity [31]. Water entering the pores and then freezing leads to the destruction of the material. European standards for natural stones methods exists (for example, EN 12371: natural stone test methods—determination of frost resistance; EN 12370: natural stone test methods—determination of resistance to salt crystallization). However, to study the impact of gradual climate change, which exacerbates the usually slow rate of degradation of outdoor materials [32,33], indicators that take into account air temperature, precipitation, relative humidity and wind are usually used. We used two indices for assessing frost damage, the so-called climate-based indices [33].

The first index is the number of freeze-thaw cycles (FTCs) [34]. Based on the average daily air temperature, a cycle is counted every time the temperature falls below 0 °C, provided that the previous day was not frosty.

The second index is the wet-frost index [34], which estimates the number of rainy days, immediately followed by days with an average air temperature below −1 °C. Calculated as the number of rainy days ($p > 2$ mm and T $> 0$ °C) followed immediately by days with mean temperature below $-1$ °C in a year [35]. Damage to building materials (stones) from frost occurs as a result of an increase in the volume of water in the pores or the volume of cracks during freezing. It is important to consider whether the stone is wet when the temperature drops.

Salt weathering is one of the most important degradation criteria in the historical heritage. The procedure for assessing the number of crystallization transitions—the dissolution of various salts that contribute to the destruction of stone and brick buildings, is described in detail and explained in the works [34,36–39]. In the case of the sodium chloride salt (NaCl, halite), this is estimated by counting the number of times the average daily relative humidity crosses the critical liquefaction point of 75.3% on consecutive days. We used two days [38]. Only transitions that occur with decreasing moisture and crystallization are taken into account, which is equivalent to the number of cycles [36]. To quantify the damage caused by calcium sulfate, we simply counted the number of days during which the relative humidity was above 80%. The choice of this threshold was made on the basis of a bibliographic review [40].

The index trend values were calculated using the least squares method, and their statistical significance was calculated using the Mann-Kendall parameter (95% significance level).

## 3. Results

### 3.1. Frost Damage Indices

The number of freeze-thaw cycles for the period 1961–2020 varies from 0 to 15 (Figure 2a). The maximum values of the index are observed in the north-west of the region (the area of the Baltic Sea), where the cultural heritage sites of St. Petersburg, Novgorod and Pskov are concentrated. Vandemeulebroucke et al. [33] obtained the freeze-thaw cycles for Helsinki which is located near St. Petersburg, equal to 12 for the historical period 1960–1989. In the east of the region (sites located in Tatarstan), the number of cycles reaches ten. The southern site of cultural heritage Naryn-Kala fortress (Derbent) demonstrate up to nine cycles per year. The Naryn-Kala facility is located on the coast of the Caspian Sea, in this area the ridge of the Caucasus Mountains comes as close as possible to the sea coast, and therefore, already at a short distance from the fortress, the number of freeze-thaw cycles increases to 11–12.

Most cultural heritage sites in the European part of Russia are located in an area with positive trends in the number of freeze-thaw cycles (Figure 2b). The regions in the Moscow area are characterized by a positive, statistically significant trend up to 1 day/10 years. Negative trends (decrease) in the number of freeze-thaw cycles are observed for the southern regions (the Naryn-Kala citadel in Derbent) ($p < 0.05$).

Frost damage to building stones occurs as a result of an increase in the volume of water in the pores or the volume of cracks during freezing [41]. Therefore, it is important to consider whether the stone is wet when the temperature drops.

On the territory of the European part of Russia, the wet-frost index reaches values of 7 (Figure 3a). However, for areas with cultural heritage sites, the index values are distributed fairly evenly—up to 5 days. In the coastal areas of the Caspian Seas (the Naryn-Kala site), the index has the lowest values (up to two cases). The spatial distribution of the trend coefficients is heterogeneous in the areas of the cultural heritage sites (Figure 3b). The northern regions of the European part of Russia are characterized by positive trends, but they are statistically insignificant.

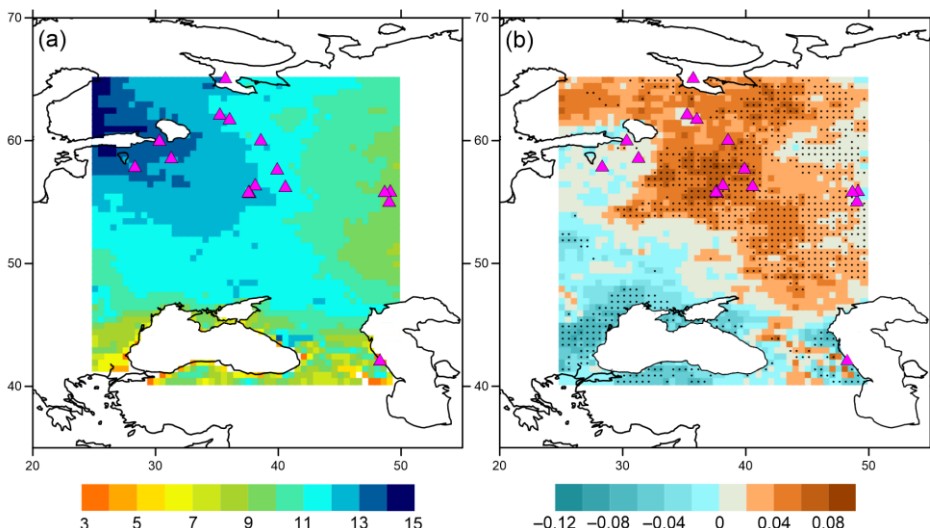

**Figure 2.** Number of FTC (**a**) and trends (**b**) for period 1961–2020. Black dots correspond to statistically significant values (*p* < 0.05). Purple triangle—cultural heritage sites.

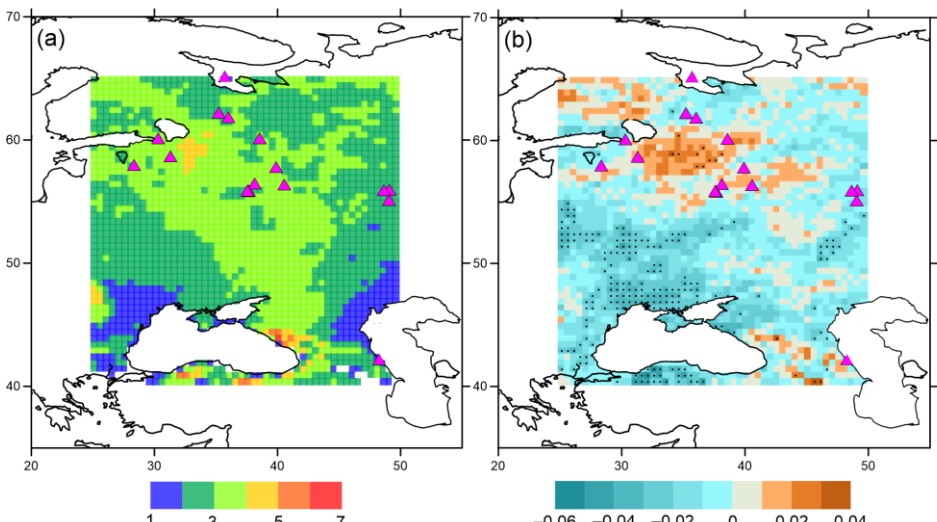

**Figure 3.** Number of WFI (**a**) and trends (**b**) for period 1961–2020. Black dots correspond to statistically significant values (*p* < 0.05). Purple triangles—cultural heritage sites.

Figure 4 shows the long-term variability of two indices for three points—St. Petersburg, Derbent and Kazan (Figure 4). For St. Petersburg, the FTC index has the greatest variation—from five to 22 freeze/thaw cycles. In the long-term course of the FTC index, inter-annual/decadal variability can be traced. For Kazan and Derbent, despite the significant distance between them, the average value of the cycles is similar (9.9 and 9.5, respectively). The value of the WFI index for a long period does not exceed 10 (Figure 4b). The average values for the three selected points differ insignificantly—3.3 in St. Petersburg, 2.8 in Derbent and 2.5 in Kazan. The average and maximum values of the frost damage indices for each cultural heritage sites are given in Table 1. As can be seen from Table 1, the maximum freeze-thaw cycles occur in 2014 in the Moscow area and its environs. The winter of that year was among the five warmest in more than 130 years of meteorological records. Frequent thaws were observed.

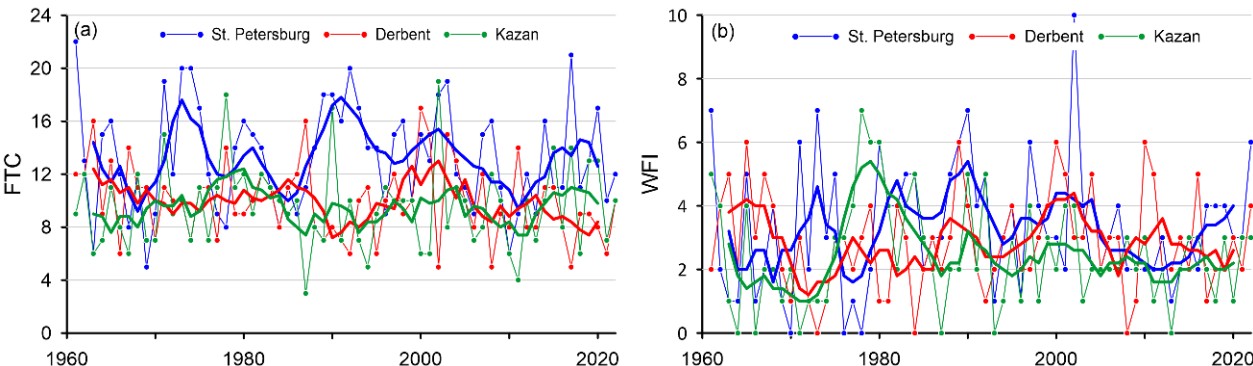

**Figure 4.** Number of FTC (**a**) and WFI (**b**) for three cities located in different climatic conditions for period 1961–2020. Bold line—5-year moving average.

**Table 1.** Average and maximum values of the frost damage indices for each cultural heritage sites.

| Cultural Heritage Sites | Freeze-Thaw Cycles | | Wet-Frost Index | |
|---|---|---|---|---|
| | Mean | Max (Year) | Mean | Max (Year) |
| Historic Centre of Saint Petersburg | 13.3 | 22/1960 | 3.3 | 10/2001 |
| Kizhi Pogost | 12.6 | 21/2018 | 2.3 | 7/1970 |
| Kremlin and Red Square | 12.0 | 21/2014 | 2.9 | 8/1991 |
| Ensemble of the Solovetsky Islands | 13.3 | 22/1970 | 1.9 | 6/2018 |
| Historic Monuments of Novgorod | 13.5 | 24/2002 | 3.4 | 8/1970 |
| White Monuments of Vladimir and Suzdal | 11.6 | 20/2014 | 3.1 | 9/1997 |
| Trinity Sergius Lavra in Sergiev Posad | 12.1 | 20/2014 | 3.5 | 8/1970 |
| Church of the Ascension | 12.0 | 21/2014 | 2.9 | 8/1991 |
| Ensemble of the Ferapontov Monastery | 11.8 | 23/2014 | 3.0 | 8/2001 |
| Kazan Kremlin | 9.9 | 19/2001 | 2.5 | 7/1977 |
| Citadel, Ancient City and Fortress Buildings of Derbent | 9.5 | 17/1999 | 2.8 | 6/1964 |
| Ensemble of the Novodevichy Convent | 12.0 | 21/2014 | 2.9 | 8/1991 |
| Historical Centre of the City of Yaroslavl | 11.7 | 21/2001 | 3.2 | 7/1977 |
| Bolgar Historical and Archaeological Complex | 9.6 | 20/2001 | 2.6 | 6/1964 |
| Assumption Cathedral and Monastery of Sviyazhsk | 9.8 | 22/2001 | 2.4 | 6/1978 |
| Churches of the Pskov School of Architecture | 12.7 | 25/1991 | 3.0 | 7/1970 |
| Petroglyphs of Lake Onega and the White Sea | 11.4 | 21/2019 | 3.1 | 7/2010 |

*3.2. Salt Crystallization*

The number of phase transitions of sodium chloride for the areas of cultural heritage sites in the territory of the European part of Russia varies from five to 40 cycles (Figure 5a). In the northwest of the study region (St. Petersburg region), the number of cycles is 20–25. For site in the south of the European part of Russia, the magnitude of the cycles is up to 25 times for the Naryn-Kala (Derbent). A statistically significant increase in the number of sodium chloride transitions was obtained for the Solovetsky Islands, as well as the St. Petersburg region (Figure 5b). Positive trends in the index are also characteristic of cultural heritage sites in the Republic of Karelia (Kizhi and Petroglyphs), Pskov and Kazan. Negative and statistically insignificant trends, however, are observed in the areas of Moscow, Vologda, Yaroslavl and Suzdal. A southern cultural heritage site is located in an area of positive (but not significant) trends.

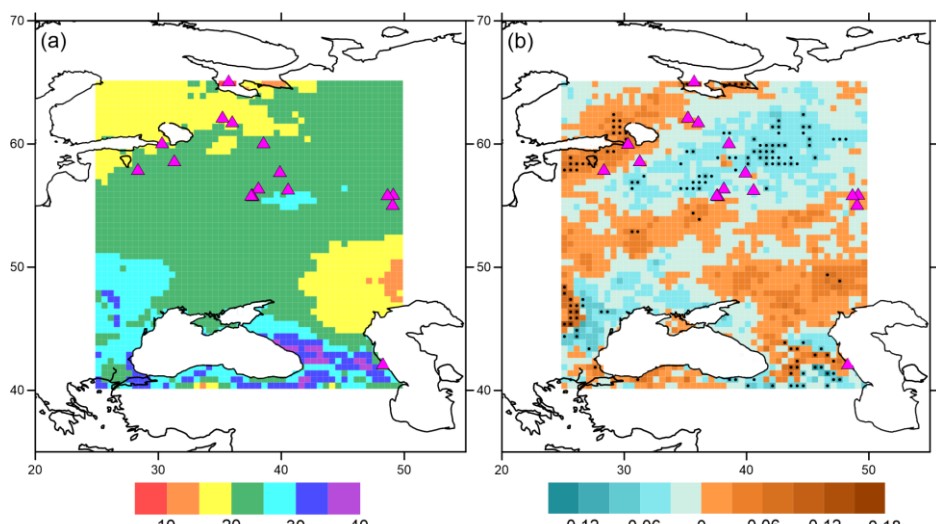

**Figure 5.** Number of salt crystallization transitions (**a**) and trends (**b**) for period 1961–2020. Black dots correspond to statistically significant values (*p* < 0.05). Purple triangles—cultural heritage sites.

The number of days with a humidity of more than 80% (damage caused by calcium sulfate) across the territory of the European part of Russia has a large range—from 50 to 300 days (Figure 6a). The largest number of days with humidity over 80% is characteristic of the Solovetsky Islands. In general, there is a decrease in the number of days with humidity over 80% in the southeast direction. The southern coastal cultural heritage site has several days with relative humidity; up to 250 days for the Naryn-Kala citadel (Derbent). In addition, the southern site is characterized by negative statistically significant trends of days with humidity above 80% (Figure 6b). A decrease in the number of days with humidity above 80% is also observed in the region of Tatarstan, Moscow and its environs (not significant). In the north-west of the region, the trend in the number of days is positive, but statistically insignificant. A statistically significant decrease in the number of days with air humidity above 80% is also characteristic of the Solovetsky Islands (Solovki).

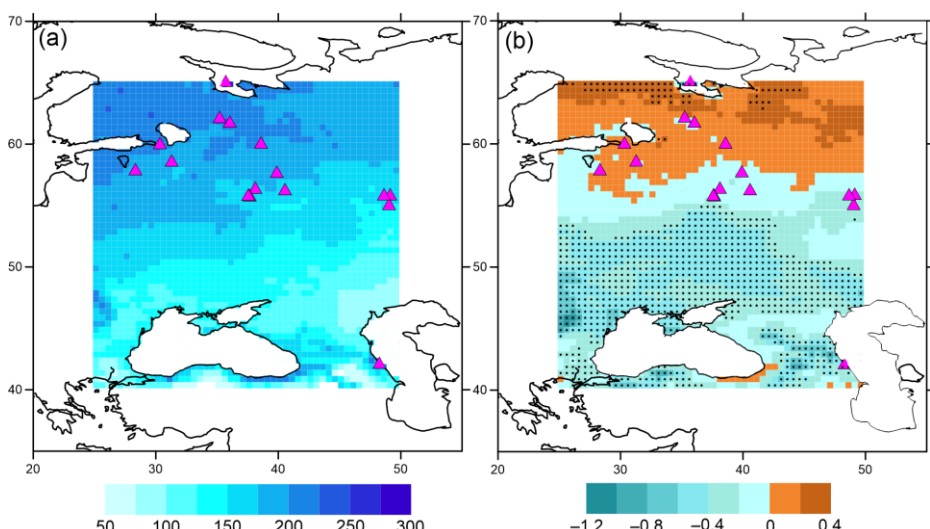

**Figure 6.** Number of days with relative humidity over 80% (**a**) and trends (**b**) for period 1961–2020. Black dots correspond to statistically significant values (*p* < 0.05). Purple triangles—cultural heritage sites.

For the selected points, the number of crystallization-dissolution transitions of sodium chloride ranges from 12 to 32 (Figure 7a). The long-term variability in Kazan and Derbent has a similar character despite the large distance between them. Possibly, longitude plays a role. In the long-term course of the number of days with relative air humidity above

80% for Derbent and Kazan, there is a pronounced negative trend, especially noticeable after 2000 (Figure 7b). For the Derbent region, the trend value reaches—6 days/10 years ($p < 0.05$). Average and maximum values of the number of crystallization-dissolution transitions of sodium chloride and the number of days with relative air humidity above 80% are presented in Table 2.

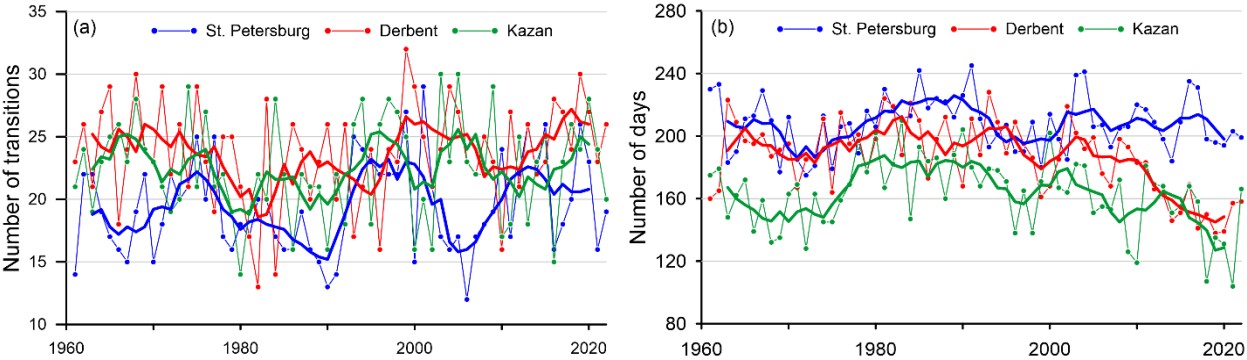

**Figure 7.** Number of salt crystallization transitions (**a**) and number of days with relative humidity over 80% (**b**) for three cities located in different climatic conditions for period 1961–2020. Bold line—5-year moving average.

**Table 2.** Average and maximum values of the number of crystallization-dissolution transitions of sodium chloride and the number of days with relative air humidity above 80% for the cultural heritage sites.

| Cultural Heritage Sites | Salt Crystallization Transitions | | Number of Days with Relative Humidity Over 80% | |
|---|---|---|---|---|
| | **Mean** | **Max (Year)** | **Mean** | **Max (Year)** |
| Historic Centre of Saint Petersburg | 19.5 | 29/2000 | 207.9 | 245/1990 |
| Kizhi Pogost | 14.2 | 23/2001 | 243.7 | 290/1973 |
| Kremlin and Red Square | 23.8 | 31/2016 | 168.9 | 201/1989 |
| Ensemble of the Solovetsky Islands | 7.8 | 15/2016 | 277.4 | 326/1960 |
| Historic Monuments of Novgorod | 20.8 | 29/1997 | 194.3 | 235/1961 |
| White Monuments of Vladimir and Suzdal | 24.0 | 31/1993 | 165.8 | 208/1979 |
| Trinity Sergius Lavra in Sergiev Posad | 23.7 | 33/1993 | 177.3 | 214/1989 |
| Church of the Ascension | 23.8 | 31/2016 | 168.9 | 201/1989 |
| Ensemble of the Ferapontov Monastery | 20.6 | 28/1993 | 196.7 | 224/1982 |
| the Kazan Kremlin | 22.5 | 30/2002 | 163.7 | 210/1982 |
| Citadel, Ancient City and Fortress Buildings of Derbent | 23.7 | 32/1998 | 187.2 | 228/1982 |
| Ensemble of the Novodevichy Convent | 23.8 | 31/2016 | 168.9 | 201/1989 |
| Historical Centre of the City of Yaroslavl | 22.9 | 31/1965 | 183.8 | 229/1989 |
| Bolgar Historical and Archaeological Complex | 22.7 | 33/1996 | 149.3 | 191/1979 |
| Assumption Cathedral and Monastery of Sviyazhsk | 22.8 | 32/1975 | 161.4 | 204/1989 |
| Churches of the Pskov School of Architecture | 21.2 | 29/1992 | 187.8 | 222/1961 |
| Petroglyphs of Lake Onega and the White Sea | 18.5 | 25/2002 | 218.0 | 255/1960 |

## 4. Discussion and Conclusions

Climate change affects all spheres of human activity, including cultural heritage sites. Under the influence of the main meteorological parameters and their extremes, the destruction (degradation) of building materials from which cultural heritage sites are created occurs. The territory of Russia is no exception. Using ERA5 reanalysis data and climate-based indices, we assessed their changes in the territory of the European part of Russia (location of UNESCO cultural heritage sites) for the period 1960–2020.

To do this, we used two climate-dependent indices for assessing the effect of frost on the destruction of building materials. The FTC index of freeze and thaw cycles is a

simple and straightforward way to assess potential damage from frost [39]. The observed widespread increase in air temperature can increase the frequency of freeze-thaw cycles associated with the passage of air temperature through 0 °C, causing damage to masonry structures [9,10,34]. For the territory of the European part of Russia, we obtained positive trends in the FTCs for almost all cultural heritage sites, except for the southern site (the Naryn-Kala citadel). An analysis of future changes based on ensemble model calculations showed that practically, throughout the European part of Russia, the number of days with the air temperature passing through 0 degrees will increase throughout the 21st century from November to March, with the largest increase in the southern regions of the European part of Russia [42]. An 81% increase in damage from freeze-thaw cycles was obtained for the north of Europe according to modeling data by the end of the 21st century (for the period 2070–2099) [33]. The second wet-frost index shows a predominant decrease in the territory of the European part of Russia. As expected, the highest values of frost-damage indices were obtained for the north of the study region, and cultural heritage sites located there are more susceptible to damage caused by frost, and these trends persist.

Salt weathering is one of the most important degradation criteria in the historical heritage, due to the phase change in relative humidity. Crystallization-dissolution transitions are very sensitive to variations in climatic parameters [43]. Climatic changes have led to an increase in dry-wet cycles and, as a result, the number of crystallization-dissolution cycles of salts in stone and brick structures is increasing [44]. Damage occurs during the cycles of crystallization and salt dissolution. An increase in salt crystallization has been found in most of Europe, especially in Central Europe [35,40]. In most parts of Norway, a higher risk of salt crystallization in the next century can probably be expected, especially in coastal areas [44]. For the French territory, a slight increase is obtained. For NaCl, the general trend is a slight increase [40]. Our results showed an increase in gypsum formation in the northern regions and a statistically significant decrease in the southern regions. An increase in the duration of liquid and mixed precipitation, changes in relative humidity and air temperature are the main factors determining the increase in the aggressiveness of the atmosphere and the intensity of destruction of structures [25]. According to the results of climate modeling, an intensification of the observed changes in the temperature and humidity regime is expected [25]. This means that destruction from the cycles of crystallization and dissolution of salts will increase in the future [40,43].

Given the observed and predicted trends in the main meteorological parameters, the detrimental destructive impact of climate change on cultural heritage sites will only increase. In view of the significant length of Russia from north to south and the difference in climatic conditions, measures for the adaptation and protection of cultural heritage sites must be adapted to local conditions and take into account the material from which the site is made.

The protection of cultural heritage objects requires the choice of measures to adapt to climate change, the application of an interdisciplinary approach [30,45], the involvement of participating states in actions to climate change, the application of new approaches using artificial intelligence [46], and remote sensing [47,48].

Review articles Daly [49], Orr et al. [50] for the period 2016–2020, Sesana et al. [32] and earlier Fatorić and Seekamp [51] for the period from 1900 to 2015 do not include any publication for the territory of Russia and sites located on it. The results of our work will help partially fill this gap and provide information on the impact of climate change on cultural heritage sites. For further research, higher resolution data will be used, as well as projections of changes in the main meteorological parameters and indices by the middle and end of the 21st century will be made.

**Author Contributions:** Conceptualization, O.S. and E.V.; methodology, O.S.; formal analysis, O.S.; investigation, E.V.; data curation, O.S.; writing—original draft preparation, E.V.; writing—review and editing, O.S. and E.V.; visualization, E.V.; supervision, E.V.; funding acquisition, E.V. All authors have read and agreed to the published version of the manuscript.

**Funding:** The study was supported by state assignment of Institute of natural and technical systems (Project Reg. No. 121122300072-3).

**Institutional Review Board Statement:** Not applicable.

**Informed Consent Statement:** Not applicable.

**Data Availability Statement:** The initial time series of daily data of the average air temperature, precipitation and dew point are on the website Climate Explorer (European Climate Assessment & Dataset) https://climexp.knmi.nl/start.cgi (accessed on 8 December 2022).

**Acknowledgments:** The authors are grateful to the editor and anonymous reviewers for the remarks and comments which led to improve the paper.

**Conflicts of Interest:** The authors declare no conflict of interest.

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
