# Peer review of "Climate Change Impact on the Cultural Heritage Sites in the European Part of Russia over the Past 60 Years"

_climate, doi:10.3390/cli11030050_

Round 1

Reviewer 1 Report

The paper aims to describe the impact of climate changes between 1960 and 2020 on 20 Russian Cultural Heritage sites listed in the UNESCO World Heritage List, through the reanalysis of three different indexes (FTCs, Wet-frost, salt weathering).

The major strenght of the paper is that it fills the gaps on climate data related to CH for the Russian territory.

Generally speaking, although the aim of the paper is based on a solid bibliography and a real (and urgent) need of safeguarding cultural heritage sites, some issues have been identified:

- While the choice and the use of the ERA5 data is well explained and methodologically correct, a factual "ground-truth" on Cultural Heritage sites is totally missing/not described: the "destructive impact of climate change con cultural heritage sites" (338-339) (that indeed is a reality) is not demonstrated directly by the data and the research described, but from references and the related a priori hypothesis. The research would benefit from ground truthing activities: in its absence, the "conclusions" should be partially rewritten, limitating the discourse on the "potential" destructive power and develop more how the collected and analysed data could help in conservative actions.

- The Results section (205-300) will benefit from tables clearly portraying the data discussed in the text for the various cultural heritage sites.

- The reference are generally good: I would suggest to develop more and eventually support with more references the thesis portrayed in lines 68-71 ("The mentioned changes certainly affect the building materials of cultural heritage sites in Russia, destroying them"). I would also suggest to take into consideration the recent review of Sesana et al. (https://doi.org/10.1002/wcc.710).

- I would avoid to use the word "object"/"objects" while referring to Cultural Heritage sites (as seen throughout the whole paper): this is a misleading word. I'd stick with the word "site"/"sites".

- In the Materials and Methods section I would see necessary an explanation on why the authors decided to limit/focus the research scope to UNESCO World Heritage sites.

The style is quite clear and well written, with just a few lines that remains unclear:

34 - "high confidence": is not clear what the authors are referring to.
65-66 - Something seems missing.
127-128 - The line is bad written or a misprint
212-213 - The authors should decide if describe the various areas in a list fashion or in a more discursive fashion. This line "Southern objects of cultural heritage" is more list-fashioned, while for the other areas the style is more discursive.
345-346 - Need to be a bit rephrased.

-

Author Response

Response to Reviewer 1

Dear Reviewer!

Thank you for your valuable comments! We tried to answer all your comments and clarify incomprehensible points. Below are the answers to your comments

The paper aims to describe the impact of climate changes between 1960 and 2020 on 20 Russian Cultural Heritage sites listed in the UNESCO World Heritage List, through the reanalysis of three different indexes (FTCs, Wet-frost, salt weathering).

The major strenght of the paper is that it fills the gaps on climate data related to CH for the Russian territory.

Generally speaking, although the aim of the paper is based on a solid bibliography and a real (and urgent) need of safeguarding cultural heritage sites, some issues have been identified:

Point 1: While the choice and the use of the ERA5 data is well explained and methodologically correct, a factual "ground-truth" on Cultural Heritage sites is totally missing/not described: the "destructive impact of climate change con cultural heritage sites" (338-339) (that indeed is a reality) is not demonstrated directly by the data and the research described, but from references and the related a priori hypothesis. The research would benefit from ground truthing activities: in its absence, the "conclusions" should be partially rewritten, limitating the discourse on the "potential" destructive power and develop more how the collected and analysed data could help in conservative actions.

Response 1: Ground truthing activities is a separate complex and lengthy work that is not included in the purpose of our work.

Point 2: The Results section (205-300) will benefit from tables clearly portraying the data discussed in the text for the various cultural heritage sites.

Response 2: The tables are added to the manuscript

Point 3: The reference are generally good: I would suggest to develop more and eventually support with more references the thesis portrayed in lines 68-71 ("The mentioned changes certainly affect the building materials of cultural heritage sites in Russia, destroying them"). I would also suggest to take into consideration the recent review of Sesana et al. (https://doi.org/10.1002/wcc.710).

Response 3: We added references. Thank you for the recommendation

Point 4: I would avoid to use the word "object"/"objects" while referring to Cultural Heritage sites (as seen throughout the whole paper): this is a misleading word. I'd stick with the word "site"/"sites".

Response 4: the word "object"/"objects" changed to "site"/"sites".

Point 5: In the Materials and Methods section I would see necessary an explanation on why the authors decided to limit/focus the research scope to UNESCO World Heritage sites.

Response 5: we have added this information to the Materials and Methods section

Point 6: The style is quite clear and well written, with just a few lines that remains unclear:

34 - "high confidence": is not clear what the authors are referring to.
65-66 - Something seems missing.
127-128 - The line is bad written or a misprint
212-213 - The authors should decide if describe the various areas in a list fashion or in a more discursive fashion. This line "Southern objects of cultural heritage" is more list-fashioned, while for the other areas the style is more discursive.
345-346 - Need to be a bit rephrased.

Response 6: edits made to the text of the manuscript. The sentences are rewritten

Reviewer 2 Report

General comments: 

-       Article is well structured and needs few minor insights:

-       English is good 

-       Some curiosities are asked to the authors and some suggestions are proposed to strengthen the outcomes of the study

Title

Instead of “Climate change impact on the cultural heritage sites in the European part of Russia”, I suggest “Climate change impact on the cultural heritage sites in the European part of Russia over the past 60 years” or, “Climate change impact on the cultural heritage sites in the European part of Russia for the period 1960-2020”

Abstract

1)     At the end of the abstract, please write just 2 or 3 sentences on the practical utility of your research for those who study the degradation of stone materials and draw up maintenance and conservation plans. You can just rearrange lines 337-342 here.

Introduction:

1)    As suggested above, strengthen the relevance of your research in the introduction after line 71. 

2)    Reviewer curiosity:

Some European standards for natural stones methods exist: 

EN 12370: Natural stone test methods - Determination of resistance to salt crystallisation

EN 14147: Natural stone test methods - Determination of resistance to ageing by salt mist

EN 12371: Natural stone test methods - Determination of frost resistance

EN 14066: Natural stone test methods - Determination of resistance to ageing by thermal shock

The standards recommend a series of cycles of salt crystallisation (30 and 60), freezing-thawing (60) and wetting-drying (20). 

Do you think that this number of cycles can be representative of the indexes you have calculated based on real climatic parameters? For you, are these standards adequate for correcting dimensioning stone claddings/covering (or any type of construction material) of modern and future new buildings? 

Authors are free to think about that, but I advise you to give a general more practical/technical contribution to the standardisation bodies and standard drafting in 5 or 6 lines. 

Materials and Methods

1)     Line 122: please modify “objects” with monuments/buildings or whatever

Results

Results are good

Discussion and conclusions

1)     What about the indexes of salt crystallisation for future ICPP in Russia? Are the indexes you have calculated supposed to change in positive or negative trend? What is your opinion? 

Author Response

Response to Reviewer 2

Dear Reviewer!

Thank you for your valuable comments! We tried to answer all your comments and clarify incomprehensible points. Below are the answers to your comments

General comments: 

-       Article is well structured and needs few minor insights:

-       English is good 

-       Some curiosities are asked to the authors and some suggestions are proposed to strengthen the outcomes of the study

Point 1: Title. 

Instead of “Climate change impact on the cultural heritage sites in the European part of Russia”, I suggest “Climate change impact on the cultural heritage sites in the European part of Russia over the past 60 years” or, “Climate change impact on the cultural heritage sites in the European part of Russia for the period 1960-2020”

Response 1: The title was corrected

Point 2: Abstract

 At the end of the abstract, please write just 2 or 3 sentences on the practical utility of your research for those who study the degradation of stone materials and draw up maintenance and conservation plans. You can just rearrange lines 337-342 here.

Response 2: we added sentences to the abstract

Point 3: Introduction:

As suggested above, strengthen the relevance of your research in the introduction after line 71. 

Response 3: we have added the information

Point 4:  Reviewer curiosity:

Some European standards for natural stones methods exist: 

 EN 12370: Natural stone test methods - Determination of resistance to salt crystallisation

EN 14147: Natural stone test methods - Determination of resistance to ageing by salt mist

EN 12371: Natural stone test methods - Determination of frost resistance

EN 14066: Natural stone test methods - Determination of resistance to ageing by thermal shock

 The standards recommend a series of cycles of salt crystallisation (30 and 60), freezing-thawing (60) and wetting-drying (20). 

Do you think that this number of cycles can be representative of the indexes you have calculated based on real climatic parameters? For you, are these standards adequate for correcting dimensioning stone claddings/covering (or any type of construction material) of modern and future new buildings? 

Authors are free to think about that, but I advise you to give a general more practical/technical contribution to the standardisation bodies and standard drafting in 5 or 6 lines. 

Response 4: The analysis of literary sources showed the use of the indexes given by us to assess the impact of climate change on cultural heritage sites in different regions of the globe. You have calculated commonly used indexes in order to compare our results with those of other authors. We have added information to the Methods section.

Point 5: Materials and Methods

Line 122: please modify “objects” with monuments/buildings or whatever

Response 5:  Corrected

Point 6: Results

Results are good

Response 6:  Thank you!

Point 7: Discussion and conclusions

What about the indexes of salt crystallisation for future ICPP in Russia? Are the indexes you have calculated supposed to change in positive or negative trend? What is your opinion? 

Response 7: we have added this information to the Discussion section
